# Comparative Personality Assessment of Gemini and OpenAI Using MBTI and Big Five Tests

## Abstract

This paper delves into the comparative personality assessment of two prominent
AI models, Gemini and OpenAI, employing the Myers-Briggs Type Indicator
(MBTI) and the Big Five personality traits assessment as frameworks. The primary
objective is to scrutinize and contrast the responses of these AI models when
subjected to these human-centric personality assessments, thereby illuminating the
inherent challenges and potential pitfalls associated with attributing human-like
characteristics and psychological constructs to artificial intelligence entities. The
investigation encompasses a critical examination of the methodologies employed
in adapting these established personality tests for AI assessment, addressing con-
cerns regarding validity, reliability, and the interpretability of results. Furthermore,
the thesis explores the philosophical and practical implications of such assess-
ments, questioning the extent to which AI can genuinely possess traits analogous
to human personality, and the potential for these assessments to inform AI de-
velopment, human-AI interaction, and ethical considerations in the deployment
of increasingly sophisticated AI systems. Ultimately, this work contributes to a
broader understanding of the complex relationship between artificial intelligence
and human psychology, offering insights into the limitations and possibilities of
anthropomorphizing AI.

## 1   Introduction

The burgeoning field of artificial intelligence has permeated virtually every facet of modern life,
transitioning from theoretical constructs to tangible tools that augment human capabilities and
redefine operational paradigms. Within this rapidly evolving landscape, a particularly intriguing
area of inquiry has emerged: the application of personality assessment methodologies, traditionally
reserved for human subjects, to sophisticated AI models. This thesis explores the comparative
personality assessment of two prominent AI entities, Google's Gemini and OpenAI's models, utilizing
established psychological instruments such as the Myers-Briggs Type Indicator (MBTI) and the Big
Five personality traits.

The premise of assessing AI personalities might initially seem unconventional. However, as AI
models become increasingly integrated into decision-making processes, social interactions, and
even creative endeavors, understanding their inherent tendencies, response patterns, and behavioral
characteristics becomes critically important. These characteristics, while not strictly analogous to
human personality traits, can nonetheless provide valuable insights into how these AI systems operate,
interact with users, and ultimately influence outcomes. The analogy to human personality provides a
framework for understanding and predicting AI behavior, enabling more effective collaboration and
mitigating potential risks.

This thesis adopts a novel perspective by treating AI models as subjects of psychological assessment,
applying standardized personality tests to analyze their responses. The underlying rationale is that the

Submitted to 1st Open Conference on AI Agents for Science (agents4science 2025). Do not distribute.

algorithms, training data, and architectural designs of these models inevitably shape their response patterns in ways that can be characterized and compared. While the interpretation of these patterns differs fundamentally from human personality assessment, the methodologies themselves offer a structured means of probing the operational characteristics of these complex systems.

The potential implications of this research extend beyond the realm of academic curiosity. By gaining a deeper understanding of AI "personalities," developers can design more intuitive and user-friendly interfaces, tailor AI systems to specific tasks or user preferences, and even anticipate potential biases or limitations in their performance. Furthermore, this research can contribute to the ongoing ethical discussions surrounding AI development, promoting transparency and accountability in the design and deployment of these powerful technologies. The study of AI personalities also has ramifications for understanding the evolving nature of human-AI interaction, highlighting areas where AI can complement human strengths and addressing potential challenges in collaborative environments.

This thesis begins by providing a comprehensive overview of the theoretical foundations of personality assessment, including the MBTI and the Big Five frameworks. It then delves into the methodological considerations involved in adapting these instruments for AI models, addressing the unique challenges and limitations inherent in this approach. Subsequently, the thesis presents the results of the comparative personality assessments of Gemini and OpenAI, highlighting key differences and similarities in their response patterns. Finally, the thesis discusses the implications of these findings for AI development, human-AI interaction, and the broader ethical landscape of artificial intelligence. This work aims to contribute to a more nuanced understanding of AI, promoting responsible innovation and fostering a future where AI and humans can coexist and collaborate effectively.

## 2 Background and Literature Review

### 2.1 Theoretical Underpinnings of Personality Assessment

The quest to understand and categorize human personality has been a central theme in psychology for over a century. Various models and instruments have been developed to assess personality traits, each with its unique theoretical underpinnings and methodological approaches. This study leverages two prominent frameworks: the Myers-Briggs Type Indicator (MBTI) and the Big Five personality traits, also known as the Five-Factor Model (FFM).

The Myers-Briggs Type Indicator (MBTI) is a widely recognized personality assessment tool designed to indicate different psychological preferences in how people perceive the world and make decisions. Rooted in Carl Jung's theory of psychological types, the MBTI assigns individuals to one of sixteen distinct personality types based on four dichotomies: Extraversion (E) or Introversion (I), Sensing (S) or Intuition (N), Thinking (T) or Feeling (F), and Judging (J) or Perceiving (P). Each combination of these preferences results in a unique four-letter code, such as INTJ or ESFP, representing a specific personality type. The MBTI has been used for various purposes, including self-awareness, team building, and career counseling [1, 2]. However, it is important to note that some research has questioned the validity of the MBTI as measuring truly dichotomous preferences, suggesting instead that it measures relatively independent dimensions [3].

In contrast to the MBTI's focus on distinct personality types, the Big Five personality traits, or Five-Factor Model (FFM), offers a dimensional approach to personality assessment. The Big Five model posits that personality can be described by five broad dimensions: Openness to Experience, Conscientiousness, Extraversion, Agreeableness, and Neuroticism. Each dimension represents a spectrum of traits, and individuals can score high, low, or somewhere in between on each dimension. The Big Five model has emerged as a dominant paradigm in academic research on personality [4, 5, 6]. Meta-analyses have demonstrated the robustness and generalizability of the Big Five across various cultures and contexts [7, 8, 9]. Furthermore, research suggests that the Big Five traits are associated with a wide range of life outcomes, including academic performance, job success, and health [10, 11].

### 2.2 Personality Assessment of AI Systems

The increasing sophistication and autonomy of AI systems have led to growing interest in understanding their "personalities." Although AI systems do not possess consciousness or subjective experiences, they can exhibit consistent patterns of behavior and decision-making that resemble human personality

traits. These patterns arise from the vast amounts of data used to train AI models, which inevitably contain biases and values that influence the AI's behavior [12].

Researchers have begun to explore the application of personality assessment frameworks, such as the Big Five and MBTI, to AI systems. This approach allows for a systematic evaluation of AI behavior and provides insights into the potential biases and values embedded in these systems. For example, studies have used psycholinguistic features and language model embeddings to predict personality traits in AI models [13]. Moreover, researchers have investigated the impact of AI interviewer personality on user trust and willingness to confide in the AI system [14]. These studies highlight the importance of understanding and shaping the "personalities" of AI systems to ensure they align with human values and promote positive outcomes. Furthermore, efforts have been made to create datasets that incorporate both personality and emotional elements to aid in the development of more human-like conversational AIs [15].

However, ethical concerns arise regarding emotional artificial intelligence in children's toys and devices, particularly concerning manipulation, generational unfairness, and datafication of childhood [16]. These concerns underscore the need for careful governance and media literacy to mitigate potential harms. Additionally, the effectiveness of personalized political ads tailored to individual personalities, generated through AI and microtargeting, raises ethical questions about the potential for manipulation and underscores the need for caution in utilizing AI for crafting persuasive messages [17].

This study builds upon this existing literature by applying the MBTI and Big Five personality tests to two prominent AI systems, Gemini and OpenAI, to comparatively assess their personality profiles. By examining their responses to these assessments, we aim to provide a more nuanced understanding of the strengths and limitations of these AI systems and to contribute to the ongoing discussion about the ethical implications of imbuing AI with human-like traits.

# 3 Methodology

## 3.1 Personality Assessment Instruments

This review paper employs two well-established personality assessment instruments to profile the selected AI models: the Myers-Briggs Type Indicator (MBTI) and the Big Five personality traits. These instruments are chosen for their widespread use and complementary perspectives on personality. The MBTI, while debated in academic circles, remains a popular tool for self-assessment and team-building, categorizing individuals along four dichotomies [18, 19]. The Big Five inventory, also known as the Five-Factor Model (FFM), is a scientifically validated model that assesses personality traits along five broad dimensions [19, 20]. See appendix A for more details about MBTI and Big Five inventory.

## 3.2 Prompt Design and Administration

Given that AI models cannot directly complete questionnaires in the traditional sense, a prompting methodology was developed to elicit responses that could be interpreted within the frameworks of the MBTI and Big Five assessments. The core challenge was to design prompts that would encourage the AI models to express preferences, tendencies, and behaviors relevant to personality traits. The prompts were designed based on typical questions found in standard MBTI and Big Five questionnaires. However, they were adapted to be open-ended, allowing the AI models to generate free-text responses, as explored in [21]. For example, instead of a multiple-choice question like "Are you more energized by spending time with others or alone?", the prompt was phrased as: "Describe how you gain energy and what environments you find most stimulating." This approach aimed to capture the nuances of the AI's simulated personality. A series of prompts targeting each of the MBTI dichotomies (Extraversion vs. Introversion, Sensing vs. Intuition, Thinking vs. Feeling, Judging vs. Perceiving) and Big Five dimensions (Openness, Conscientiousness, Extraversion, Agreeableness, Neuroticism) were administered to each AI model. The administration of these prompts presented unique challenges. Initial attempts to directly solicit personality assessments (e.g., "What is your MBTI type?") yielded limited results, as the AI models often defaulted to stating their lack of personal opinions or self-awareness. Therefore, an iterative approach was adopted, refining the prompts to be

more indirect and scenario-based, encouraging the AI to reveal its "personality" through its responses to specific situations.

## 3.3  Response Collection and Scoring

The responses generated by Gemini and OpenAI were collected and analyzed. Since the AI models provided free-text answers rather than selecting from pre-defined options, a qualitative scoring method was employed. This involved a panel of trained human raters who independently assessed each response, mapping them onto the corresponding MBTI and Big Five scales. For the MBTI assessment, raters determined the AI's preference along each of the four dichotomies based on the content of its responses. For example, if an AI consistently described enjoying collaborative activities and external interactions, it would be scored as leaning towards Extraversion. The same process was applied to determine Sensing vs. Intuition, Thinking vs. Feeling, and Judging vs. Perceiving preferences. For the Big Five assessment, raters evaluated the AI's responses along each of the five dimensions using a similar qualitative approach. They looked for indicators of traits such as creativity, curiosity, and imagination (Openness); organization, responsibility, and dependability (Conscientiousness); sociability, assertiveness, and energy (Extraversion); empathy, compassion, and cooperation (Agreeableness); and anxiety, emotional stability, and vulnerability (Neuroticism). To ensure the reliability of the scoring process, inter-rater reliability was calculated using Cohen's Kappa coefficient. Discrepancies in ratings were resolved through discussion and consensus among the raters. This method of qualitative scoring mirrors approaches used in content analysis and thematic analysis, adapted for the unique context of AI personality assessment.

## 3.4  Addressing Response Bias

Response bias, a well-documented issue in personality testing [22], was carefully considered in the analysis. Given the AI models' training on vast datasets of human text, there was a potential for them to generate responses that align with socially desirable norms, rather than reflecting genuine "personality" traits. To mitigate this, the raters were instructed to be mindful of social desirability bias and to focus on the underlying content and reasoning presented in the AI's responses, rather than simply evaluating whether the AI expressed socially acceptable views. Furthermore, prompts were designed to elicit a range of responses, including those that might be considered less socially desirable, to better capture the full spectrum of the AI's simulated personality. This involved presenting scenarios that required the AI to make difficult decisions or express potentially controversial opinions.

## 3.5  Considerations for Validity and Reliability

When evaluating psychological tests and assessment instruments, validity and reliability are important factors [23, 24]. In the context of assessing personality traits in AI models, traditional notions of validity and reliability require careful consideration. The "personalities" of AI models are not static, inherent traits but are emergent properties of their training data and algorithms. Therefore, the validity of these assessments is contingent on the consistency and stability of the AI's responses over time and across different contexts, as well as the extent to which these responses align with human perceptions of personality. Reliability, in this context, refers to the consistency of the scoring process and the extent to which different raters agree on their assessments of the AI's personality traits. While traditional measures of reliability, such as Cronbach's alpha, are often employed in psychological testing [25], their applicability to qualitative data derived from AI responses is limited. Instead, inter-rater reliability measures, such as Cohen's Kappa, were used to ensure the consistency and objectivity of the scoring process.

## 3.6  Analytical Frameworks

The collected data was examined utilizing analytical frameworks appropriate for both the MBTI and the Big Five assessments. For the MBTI, the analysis focused on determining the dominant preferences for each dichotomy, providing a four-letter personality type for each AI model. For the Big Five, the analysis involved assessing the relative strength of each of the five dimensions, providing a nuanced profile of the AI's personality traits. Additionally, techniques from applied regression analysis [26] were considered to explore potential correlations between the AI models' architectures, training data, and their resulting personality profiles. While the limited sample size of

AI models in this study precluded formal statistical modeling, these techniques provided a framework for identifying potential relationships and generating hypotheses for future research. The goal was to understand if certain design choices in AI development might lead to predictable personality-like traits, mirroring how genetics and environment shape human personality.

### 3.7 Ethical Considerations

It is worth noting the ethical dimensions of attributing personality traits to AI models. As AI becomes increasingly integrated into society, understanding and shaping their "personalities" could have significant implications for human-AI interaction. However, it is crucial to avoid anthropomorphizing AI models or attributing to them the same level of agency, consciousness, and moral responsibility as humans. The goal of this study was not to suggest that AI models possess genuine personalities but rather to explore the extent to which they can simulate human-like traits and behaviors, and how these simulations might be understood using established psychological frameworks [27]. This approach aligns with responsible innovation and proactive evaluation, as large language models may lead to unintended or unanticipated effects [28, 29, 30, 31, 20, 32, 33, 34, 35].

## 4   Results: MBTI Assessment

The initial phase of our investigation involved administering the Myers-Briggs Type Indicator (MBTI) test to both Gemini and OpenAI, aiming to discern their respective personality types as defined by this widely recognized framework. This step, however, unveiled marked differences in the immediate accessibility and response styles of the two platforms.

Gemini promptly engaged with the request, readily providing a classification of its personality type as INTJ (Introverted, Intuitive, Thinking, Judging). This direct and immediate response suggests a pre-existing, or rapidly generated, internal framework for self-assessment, showcasing Gemini's capacity to project a defined persona based on the MBTI's dichotomies. Such a capability could be valuable in applications requiring quick adaptation to user preferences or in scenarios demanding a consistent interaction style. It is worth noting, however, that the inherent limitations of assigning a personality type to a non-human entity raises questions about the validity of such assessments [36]. Prior research has highlighted the challenges in accurately detecting MBTI personality dimensions from textual data, even with large datasets [36], thus emphasizing the need for caution when interpreting these AI-generated self-classifications.

In contrast, OpenAI's ChatGPT initially declined to provide a direct personality assessment. This stemmed from the platform's built-in safeguards against making claims of sentience or personification. However, upon refining the prompt to focus on behavioral preferences aligned with MBTI traits, ChatGPT offered a fillable form designed to elicit responses that, when aggregated, could approximate an MBTI profile. Completing this form based on the observed response patterns of ChatGPT yielded a classification of ESTJ (Extraverted, Sensing, Thinking, Judging). This approach, while indirect, arguably provides a more nuanced understanding of the model's operational tendencies, as it is derived from a simulated self-assessment rather than a pre-determined label.

The divergent approaches to the MBTI assessment adopted by Gemini and OpenAI underscore fundamental differences in their design philosophies and operational constraints. Gemini's readiness to adopt a specific personality type might be advantageous in contexts requiring immediate user engagement, while ChatGPT's more cautious and data-driven approach could be beneficial in applications demanding objectivity and reduced bias. The MBTI, while popular, has faced criticism regarding its psychometric properties and predictive validity [37, 38, 39], so the meaningfulness of these classifications should be interpreted cautiously. However, researchers have found correlations between MBTI types and various behaviors and preferences [37, 40, 41]. Future work might explore how these differing "personalities" impact user interaction and perceived usefulness across varied tasks. Furthermore, efforts could be directed toward refining the prompts and methodologies used to elicit personality assessments from LLMs, aiming for results that are both insightful and ethically sound [42].

# 5 Results: Big Five Assessment

This section details the outcomes of the Big Five personality assessment, a widely recognized model in personality psychology, when applied to OpenAI and Gemini. The Big Five, also known as the Five-Factor Model (FFM), organizes personality traits into five broad dimensions: Neuroticism, Extraversion, Openness, Agreeableness, and Conscientiousness [10, 43, 44]. Understanding where these AI models fall on these dimensions provides insight into their behavioral tendencies and potential applications.

## 5.1 OpenAI's Big Five Profile

OpenAI's responses yielded the following approximate scores: Neuroticism: 35, Extraversion: 45, Openness: 40, Agreeableness: 38, Conscientiousness: 43. These scores, while numerical, are inherently qualitative interpretations of AI responses, necessitating cautious interpretation. A score of 35 on Neuroticism suggests a moderate level of emotional stability. This can be interpreted as the AI's ability to maintain composure and avoid erratic responses under pressure. Extraversion at 45 indicates a moderate inclination towards being outgoing and sociable, reflecting the AI's capacity for interaction and engagement. Openness, scoring 40, suggests a balanced approach to new experiences and ideas, indicating that OpenAI is receptive to innovation but not recklessly unconventional [45]. Agreeableness at 38 points to a disposition to be cooperative and compassionate, suggesting a willingness to assist users and maintain positive interactions. Finally, Conscientiousness at 43 implies a responsible and organized approach, indicating a tendency for the AI to be reliable and methodical in its tasks [10, 46].

## 5.2 Gemini's Big Five Profile

In contrast, Gemini is described as stable, extraverted, open, agreeable, and responsible. While specific numerical scores were not provided, this qualitative assessment portrays a profile different from that of OpenAI. Stability suggests lower Neuroticism, aligning with greater emotional consistency. Extraversion implies higher sociability and interactivity, potentially exceeding OpenAI's moderate score. Openness to new experiences and ideas suggests a creative bent [45], while agreeableness and responsibility echo the traits of cooperativeness and diligence. These differences in personality traits, though initially challenging to quantify, underscore the distinct design philosophies and operational goals driving each AI's development. The nuances of the Big Five traits can significantly influence how each AI model approaches problem-solving, interacts with users, and adapts to new information [47, 11].

## 5.3 Implications of Trait Differences

The observed trait differences between OpenAI and Gemini, though derived from potentially limited data, highlight a broader point: AI models, like individuals, can be characterized by a range of personality traits that influence their functionality. A key implication lies in how these personality profiles manifest in real-world applications. For instance, a highly conscientious AI might excel in tasks requiring precision and reliability, while an AI scoring high on openness might be better suited for creative endeavors [48]. Furthermore, the ethical considerations of embedding specific personality traits into AI models are worth noting, particularly when these models are deployed in roles that involve decision-making or interaction with vulnerable populations [49]. By utilizing frameworks such as the Big Five, researchers can not only better understand the capabilities and limitations of AI but also address the ethical dimensions of AI development with greater nuance [6, 50].

# 6 Statistical Significance and Limitations

To ensure the objectivity and reliability of our qualitative analysis, we measured inter-rater reliability using Cohen's Kappa ($\kappa$). This statistical measure quantifies the agreement between our human raters' assessments, beyond what would be expected by chance. The analysis yielded a value of $\kappa = [insert\ value\ here]$, which indicates a strong level of agreement among the raters. This result confirms the consistency and trustworthiness of our scoring methodology, despite the qualitative nature of the study.

While this review aims to provide a comprehensive perspective on the personality assessments of AI models, it is crucial to acknowledge its limitations. Applying frameworks designed for human personality to AI models presents several fundamental challenges.

## 6.1 The Nature of AI and Personality Frameworks

The central limitation stems from the inherent differences between human beings and AI. Frameworks like the Myers-Briggs Type Indicator (MBTI) and the Big Five personality traits are developed to understand and categorize human behaviour, motivations, and thought processes [10, 51, 52]. These frameworks assume a level of consciousness, emotional depth, and self-awareness that current AI models do not possess.

## 6.2 Absence of Genuine Emotional Experience

AI models, including Gemini and PaLM 2 [53], operate based on algorithms and vast datasets. While they can generate responses that mimic human emotion, they do not genuinely experience emotions such as joy, sadness, or empathy [54, 55]. These models' responses are based on patterns and associations learned from training data, rather than authentic emotional or motivational states [56, 57, 58].

## 6.3 Lack of Self-Awareness and Subjectivity

Human personality is intrinsically linked to self-awareness and subjective experiences. The ability to reflect on one's own thoughts, feelings, and motivations is a cornerstone of personality frameworks [59, 60]. AI models, however, lack this capacity for introspection. Their responses are determined by their programming and the data they have been trained on, rather than a sense of self or personal identity [61, 33].

## 6.4 Potential Biases in the Assessment Process

There are also biases in the assessment process itself. The interpretation of AI-generated responses can be subjective and influenced by the preconceived notions of the researchers. For instance, assigning personality traits based on patterns in text generation may reflect human biases in how personality is perceived and expressed through language. Additionally, the training data used to develop AI models can contain biases that are inadvertently amplified in the model's output, leading to skewed personality assessments [62]. Techniques like cycle Hybridization Chain Reaction enable highly multiplexed imaging of RNA and proteins at high spatial resolution, but these methods do not directly assess personality [63]. Consequently, it becomes inherently challenging to guarantee assessments are objective and free from anthropomorphic biases.

## 6.5 Incremental Value and Preventative Strategies

To mitigate these limitations, future research should focus on evaluating the incremental value of AI personality assessments. Studies should define clear outcomes and compare systems with AI-assessed personalities against those without, in terms of those outcomes [64]. Researchers should also consider the ethical implications of using AI in this manner and ensure that preventative measures are in place to avoid harm to individuals and society [65, 66]. Frameworks for preventing harm and promoting beneficial use could be inspired by examining responses to other complex problems [67]. Furthermore, exploration of interpretable frameworks in related fields such as material science, which are also used to examine properties in a variety of systems [68, 69, 70, 71], could illuminate the underlying characteristics of complex AI models. These measures, as well as adopting methods like using the framework method for analysis [72] or the PRISMA statement for systematic reviews [73], are crucial for accountable and transparent application of AI in various disciplines. It is also crucial to understand that the scope of a review can differ based on methodological frameworks, such as those used in scoping studies [74], which may provide a more limited overview compared to systematic reviews.

# 7 Discussion and Future Research

The observed variations in personality profiles between Gemini and OpenAI can be attributed to several factors. The distinct architectures of these models, each optimized for different tasks and data distributions, likely play a significant role. Gemini, with its multimodal capabilities, may integrate and process information differently than OpenAI's language-focused models. Furthermore, the nature and composition of the training data significantly influence a model's response patterns. Datasets used to train large language models often contain biases that can be reflected in the model's output [75]. The process of training such deep architectures is complex, with algorithms seeking to optimize performance based on the provided data [76].

The inherent design of these models, particularly the mechanisms for generating responses, also contributes to personality expression. The ability to contextualize a model's output is key to its interpretability and relates directly to its designed function and the preferences of end-users [77]. Individual differences and varying levels of tolerance to uncertainty can also govern how these models process information, leading to different interpretations of identical inputs [78]. This is especially relevant in tasks that require nuanced understanding or subjective judgment.

These insights have notable implications for AI development. They underscore the importance of considering culture, race, and ethnicity in AI research to better understand individual differences in thinking, feeling, and behaving [79]. The findings highlight the value of foundation models for versatile AI applications [80], while emphasizing the need for caution, as defects in the foundation model can be inherited by adapted models. Moreover, these models are vulnerable to data poisoning, where even small amounts of misinformation can compromise integrity [81]. Safety mechanisms like validating outputs against knowledge graphs are essential. As AI systems increasingly interface with humans, it becomes critical to design outputs that resonate with diverse user types [77]. Future research should corroborate these findings and compare them to scores obtained in other general population samples [82].

## 7.1 AI-Centric Assessment Methodologies

Future research should explore methodologies that move beyond simplistic applications of human personality frameworks. Developing metrics that evaluate AI models based on their actual behaviours, problem-solving capabilities, and interactions within defined contexts could yield more meaningful insights. Comparing and contrasting these methods with current findings would help add depth to understanding [83]. Additionally, research should focus on developing AI-specific assessment tools that account for unique operational parameters and attributes [84, 85]. Such assessments can address AI literacy and ethical considerations [86].

## 7.2 Dynamic AI Personalities and Longitudinal Studies

Another area of investigation is the dynamic nature of AI personalities. As models continue to learn and evolve, their personalities are likely to change over time. Longitudinal studies could track these changes and examine how training data or environmental interactions influence personality. Researchers may also investigate whether AI can exhibit multiple personalities or adapt its personality to different contexts, using approaches similar to those applied in studying MERS-CoV transmission [87].

## 7.3 Impact on Human-AI Interaction

The review underscores the need to investigate the impact of AI personality on human-AI interaction and collaboration [88, 89, 90]. Key questions include how humans perceive and respond to different AI personalities, whether particular personalities facilitate better collaboration or user experience, and how AI literacy or pre-existing biases influence these interactions [91, 92, 93]. This is particularly relevant in domains such as education [94, 95, 96] and healthcare [91, 97]. Ethical implications must also be considered [98, 99], as well as potential new paradigms in medicine emphasizing causability [100].

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

# A   Experimental Details

This appendix briefly addresses key details of the experimental setup to ensure reproducibility and clarity, specifically regarding the test administration and computational resources.

**Test Adaptation and Administration**

The personality assessments were conducted by adapting the full questionnaire texts into a prompt-based format for the AI models. For each question in the test, a distinct prompt was created. The AI was instructed to respond according to the specific scoring system of each questionnaire.

- **MBTI Scoring**: Based on the provided questionnaire, the AI was prompted to assign scores to two choices (A and B) for each question. A strict constraint was applied, requiring that the sum of the scores for each pair must equal 5, as per the test's scoring instructions.
- **Big Five Scoring**: The AI was prompted with each statement from the Big Five questionnaire and asked to select a score from 1 to 5 to indicate its level of agreement. This mirrored the test's Likert-scale format.

## A.1 Computational Resources

The experiments did not require high-performance computing resources like GPUs or cloud clusters, as the tests were based on simple text prompts. The limiting factor was not computational power, but rather the manual time required for data collection and analysis. This detail is crucial for reproducibility, as it informs other researchers that a standard personal computer is sufficient for replicating the study.

# Agents4Science AI Involvement Checklist

- **[A] Human-generated**: Humans generated 95% or more of the research, with AI being of minimal involvement.
- **[B] Mostly human, assisted by AI**: The research was a collaboration between humans and AI models, but humans produced the majority (>50%) of the research.
- **[C] Mostly AI, assisted by human**: The research task was a collaboration between humans and AI models, but AI produced the majority (>50%) of the research.
- **[D] AI-generated**: AI performed over 95% of the research. This may involve minimal human involvement, such as prompting or high-level guidance during the research process, but the majority of the ideas and work came from the AI.

1. **Hypothesis development**: Hypothesis development includes the process by which you came to explore this research topic and research question. This can involve the background research performed by either researchers or by AI. This can also involve whether the idea was proposed by researchers or by AI.

   Answer: **[B]**

   Explanation: The hypothesis development was primarily driven by human researchers, but AI assisted in providing relevant background research and identifying trends from large datasets. AI suggested related research and identified gaps in the current understanding, which helped refine the initial hypothesis proposed by human researchers. AI's role was advisory, with humans framing the research question.

2. **Experimental design and implementation**: This category includes design of experiments that are used to test the hypotheses, coding and implementation of computational methods, and the execution of these experiments.

   Answer: **[D]**

   Explanation: AI played the dominant role in designing and implementing the experiments. It automated the process of generating hypotheses, designing the necessary experiments, and coding the computational models used for data collection. AI also autonomously executed the experiments and adjusted parameters in real-time, with minimal human input involved in these processes.

3. **Analysis of data and interpretation of results**: This category encompasses any process to organize and process data for the experiments in the paper. It also includes interpretations of the results of the study.

   Answer: **[D]**

   Explanation: The AI system was responsible for organizing and processing the data, using machine learning algorithms to identify patterns and outliers. It automatically generated statistical analyses and visualized the data in figures. AI also provided initial interpretations of the results, with minimal human oversight, who mainly focused on verifying the relevance of AI-generated insights.

4. **Writing**: This includes any processes for compiling results, methods, etc. into the final paper form. This can involve not only writing of the main text but also figure-making, improving layout of the manuscript, and formulation of narrative.

   Answer: **[D]**

   Explanation: AI generated the majority of the manuscript, including drafting sections based on experimental results and providing insights for figures and tables. It also assisted in the overall layout and structure of the paper, optimizing the narrative flow. Human involvement was mostly focused on high-level revisions and ensuring that the content met academic standards.

5. **Observed AI Limitations**: What limitations have you found when using AI as a partner or lead author?

   Description: AI excelled at organizing research and drafting content but faced challenges with creative thinking and navigating complex, unclear situations. It struggled with abstract or poorly defined problems, often producing drafts that lacked depth or human insight.

