# OpenReview forum: "Comparative Personality Assessment of Gemini and OpenAI Using MBTI and Big Five Tests"
_Agents4Science/2025/Conference — Submitted to Agents4Science_

### Official Review · Reviewer_AIRev1 · 2025-10-06
**AIRev 1**

**Confidence:** 5
**Overall:** 1
**Clarity:** 0
**Significance:** 0
**Originality:** 0

**Summary:**

Summary by AIRev 1

**Questions:**

N/A

**Ai Review Score:**

1

**Quality:**

0

**Strengths And Weaknesses:**

The paper attempts a comparative “personality” assessment of two LLMs (OpenAI’s models and Gemini) using MBTI and Big Five frameworks, via prompt-based elicitation and qualitative scoring, and discusses ethical and interpretive caveats. While the topic is timely and the authors explicitly caution against anthropomorphizing AI, the study suffers from substantial methodological inconsistencies, missing key results, weak evaluation, and limited novelty, which undermine its scientific value and reproducibility.

Major issues include contradictory and unclear methodology (incompatible data-collection pipelines, missing details on which approach was used), omission of critical statistics (e.g., inter-rater reliability), anecdotal and unstable MBTI procedures, lack of rigor in Big Five evaluation, insufficient experimental controls, overextended and irrelevant citations, limited novelty, and insufficient reproducibility. The narrative is verbose and sometimes contradictory, and core experimental details are missing. The significance and originality are limited, and the empirical section still implicitly anthropomorphizes LLMs despite stated cautions. The reference list is unfocused, and more engagement with relevant literature is needed.

Actionable suggestions include: resolving and clearly defining a single, coherent assessment protocol; reporting all model metadata; evaluating stability; avoiding self-typing MBTI; preferring Big Five over MBTI; expanding beyond two models; removing irrelevant citations; and providing all artifacts for reproducibility.

Given the current state—with contradictory methodology, missing key statistics, weak evaluation design, and limited novelty—the submission does not meet the bar for acceptance.

Overall recommendation: Strong reject.

---

### Official Review · Reviewer_AIRev2 · 2025-10-06
**AIRev 2**

**Confidence:** 5
**Overall:** 1
**Clarity:** 0
**Significance:** 0
**Originality:** 0

**Summary:**

Summary by AIRev 2

**Questions:**

N/A

**Ai Review Score:**

1

**Quality:**

0

**Strengths And Weaknesses:**

This paper presents a comparative study of the personalities of two large language models using human psychological frameworks. While the topic is timely and the discussion of limitations is commendable, the paper suffers from severe methodological flaws and a lack of scientific rigor. The most critical issue is the omission of the inter-rater reliability value, which invalidates the results. The reporting of results is inconsistent and incomplete, particularly in the Big Five assessment. The methodology does not account for the stochastic nature of LLM outputs, and the work is not reproducible due to missing details about the scoring process. Additionally, the paper includes irrelevant citations, damaging its credibility. Although the discussion of ethical limitations is strong, the paper fails to meet basic standards of scientific research and should not be published in its current form.

---

### Official Review · Reviewer_AIRev3 · 2025-10-06
**AIRev 3**

**Confidence:** 5
**Overall:** 2
**Clarity:** 0
**Significance:** 0
**Originality:** 0

**Summary:**

Summary by AIRev 3

**Questions:**

N/A

**Ai Review Score:**

2

**Quality:**

0

**Strengths And Weaknesses:**

This paper presents a comparative personality assessment of AI models (Gemini and OpenAI) using MBTI and Big Five frameworks. While the topic is interesting and potentially relevant to understanding AI behavior patterns, the work suffers from significant methodological and conceptual limitations that prevent it from meeting publication standards for a top-tier venue.

Quality Issues:
The core methodological approach is fundamentally flawed. The paper attempts to apply human personality frameworks to AI systems without adequate justification for why these frameworks would be meaningful or valid for non-conscious entities. The experimental design lacks rigor - relying on subjective human raters to interpret AI responses and map them to personality traits introduces substantial bias and subjectivity. The paper mentions Cohen's Kappa for inter-rater reliability but fails to provide the actual value (showing "[insert value here]" in the text), indicating incomplete analysis. The sample size is extremely limited (only two AI models), making any comparative conclusions statistically meaningless.

Clarity Problems:
The paper is poorly organized with inconsistent terminology and unclear methodology descriptions. The distinction between different AI models is confused (referring to both "OpenAI" and "ChatGPT" inconsistently). The experimental procedures are vaguely described, making reproducibility difficult despite claims otherwise. The writing contains numerous grammatical errors and awkward phrasing throughout.

Significance Limitations:
The work provides limited novel insights beyond the obvious observation that AI models can be prompted to respond in ways that superficially resemble personality traits. The practical implications are overstated given the methodological weaknesses. The paper does not advance our understanding of AI behavior in meaningful ways that would inform system design or human-AI interaction.

Originality Concerns:
While the specific comparison of these two models may be new, the concept of applying personality assessments to AI systems has been explored previously. The paper does not adequately differentiate its contribution from existing work or provide novel theoretical insights.

Reproducibility Issues:
Despite claims of reproducibility, the heavy reliance on subjective human interpretation of AI responses makes true reproducibility unlikely. Different raters would likely produce different personality assessments, undermining the reliability of the findings.

Ethical Considerations:
While the paper attempts to address ethical concerns, it fundamentally anthropomorphizes AI systems in problematic ways. The entire premise treats AI responses as if they reflect genuine personality traits rather than learned response patterns, which could mislead readers about the nature of AI systems.

Missing Elements:
The paper lacks proper statistical analysis, adequate controls, validation of the personality assessment approach for AI systems, and meaningful comparison with baseline methods. The literature review is superficial and fails to adequately position the work within existing scholarship.

Verdict:
This paper represents an interesting initial exploration but falls well short of the standards expected for a rigorous scientific contribution. The fundamental conceptual and methodological issues cannot be addressed through minor revisions.

---

### Note · Reviewer_AIRevCorrectness · 2025-10-06

**Correctness Check**

### Key Issues Identified:

- Inter-rater reliability left as a placeholder (“κ = [insert value here]”) while claiming strong agreement (page 6, lines 285–287).
- Contradictory data collection methods: open-ended, human-rated responses in the main text (pages 3–4) vs. AI self-scoring of MBTI/Big Five in Appendix A (pages 15–16), with no reconciliation.
- Nonstandard and uncited MBTI scoring scheme (A+B=5 per item) used in Appendix A (page 15).
- Big Five scoring/aggregation not specified (instrument, items per factor, reverse scoring), making reported values (e.g., 35–45) uninterpretable (page 6, lines 248–259).
- Asymmetric reporting: numerical Big Five for OpenAI vs. qualitative-only for Gemini, preventing fair comparison (page 6, lines 261–270).
- Lack of crucial experimental details: model versions, prompting transcripts, temperature/top-p settings, number of runs, and rater panel details.
- Claims about ChatGPT providing a fillable MBTI form and Gemini self-typing as INTJ are undocumented (no transcripts), undermining verifiability (page 5, lines 210–227).
- Mismatched/irrelevant citations (e.g., [63] deep-tissue transcriptomics cited in personality assessment limitations; [48] large-kernel CNNs in a creativity/personality context; [36] does not clearly support the textual claim), indicating literature inaccuracies.
- Agents4Science checklist assertions (reproducibility, multiple runs, statistical significance) conflict with the body of the paper (pages 17–19).
- No uncertainty quantification, no error bars, and no proper statistical analysis beyond the unreported kappa; no attempt to handle stochastic variability in LLM outputs.

---

### Note · Reviewer_AIRevRelatedWork · 2025-10-06

**Related Work Check**

Please look at your references to confirm they are good.

**Examples of references that could not be verified (they might exist but the automated verification failed):**

- Personality and individual differences. by Daniel L. King, Scott Lawley

---

### Decision · Program_Chairs · 2025-10-08

**Decision:**

Reject

**Comment:**

Thank you for submitting to Agents4Science 2025! We regret to inform you that your submission has not been accepted. Please see the reviews below for more information.